# The Relationship Between Physical Activity, Sedentary Behavior, and Subjective Well-Being: Gender Differences Among Slovak University Students

**DOI:** 10.3390/healthcare13243274

**Published:** 2025-12-13

**Authors:** Alena Buková, Justyna Krzepota, Dorota Sadowska, Tatiana Kimáková, Petra Melicharová

**Affiliations:** 1Institute of Physical Education and Sport, Pavol Jozef Šafárik University in Košice, 040 11 Košice, Slovakia; alena.bukova@upjs.sk; 2Institute of Physical Culture Sciences, University of Szczecin, al. Piastów 40B, Blok 6, 71-065 Szczecin, Poland; justyna.krzepota@usz.edu.pl; 3Department of Physiology, Institute for Sport—National Research Institute, ul. Trylogii 2/16, 01-982 Warsaw, Poland; dorota.sadowska@insp.waw.pl; 4Faculty of Medicine, Pavol Jozef Šafárik University in Košice, 040 11 Košice, Slovakia; tatiana.kimakova@upjs.sk

**Keywords:** mental health, quality of life, sedentary behavior

## Abstract

**Highlights:**

**What are the main findings?**
University students with higher levels of physical activity reported significantly better subjective well-being, including fewer somatic complaints, lower depressive mood, and higher self-esteem, with clear differences between genders.While higher physical activity levels were generally associated with more favorable well-being outcomes, a plateau pattern emerged among men, whereas among women, moderate physical activity showed a paradoxical inverse association with self-esteem, suggesting that these patterns may relate to broader motivational and gender-specific contextual factors rather than direct effects.

**What are the implications of the main findings?**
Higher physical activity levels in university populations are associated with more favorable mental health and quality-of-life outcomes; however, approaches to supporting student well-being should consider gender differences and motivational contexts.University wellness programs should address both physical inactivity and excessive sedentary behavior, focusing especially on reducing passive screen time and fostering positive, intrinsic exercise motives.

**Abstract:**

**Background/Objectives:** University students are a population vulnerable to psychological distress due to academic and lifestyle transitions. This study examined the relationships between physical activity (PA), sedentary behavior, and subjective well-being among Slovak university students, with attention to gender-specific and non-linear patterns. **Methods:** A total of 1314 first-year students (69.5% women; mean age = 20.7 ± 1.4 years) completed the IPAQ-Short Form and the Bern Questionnaire on Subjective Well-Being (BSW/A). PA levels were categorized as low, moderate, or high according to standard MET thresholds. Group differences were analyzed using nonparametric tests with Benjamini–Hochberg FDR correction. **Results:** Higher PA levels were associated with more favorable well-being outcomes, particularly higher self-esteem and joy of life and lower depressed mood and somatic complaints. Effect sizes ranged from small to moderate. Gender-specific patterns emerged: among men, well-being indicators tended to plateau beyond moderate PA, whereas among women, moderate PA showed an inverse association with self-esteem despite slightly higher median scores in the moderate-activity group. Sedentary time showed weak and mostly non-significant associations after FDR correction. **Conclusions:** Physical activity was positively associated with subjective well-being in a non-linear, gender-dependent manner. These findings suggest that approaches to supporting student well-being may benefit from considering gender differences, individual activity patterns, and motivational context.

## 1. Introduction

University students constitute a population characterized by elevated exposure to psychosocial and environmental stressors that may adversely affect mental health, emotional stability, and subjective well-being.

The transition from secondary to tertiary education represents a period of substantial developmental, social, and behavioral reorganization, typically accompanied by increased academic pressure, altered daily routines, and relative autonomy from parental and institutional support structures. Empirical evidence indicates that these contextual and behavioral shifts are associated with higher vulnerability to psychological distress and the establishment of long-term health-related habits [1,2,3]. Because lifestyle patterns adopted during early adulthood often persist into later life, the university population represents a critical target for the investigation of behavioral correlates of mental well-being and preventive health promotion [3,4].

Within the behavioral domain, physical activity (PA) has consistently been identified as a key correlate of physical and psychological health outcomes. Accumulating evidence demonstrates that higher PA levels are associated with lower stress and anxiety, enhanced affective regulation, and greater life satisfaction [5,6,7]. Conversely, sedentary behavior (SB), defined as any waking behavior characterized by an energy expenditure ≤ 1.5 METs while sitting, reclining, or lying down, has emerged as an independent correlate of psychological and physiological dysregulation [8,9]. However, the strength and direction of these associations remain heterogeneous within university samples, frequently moderated by gender, activity intensity, motivational orientation, and contextual factors such as study field or environmental conditions [10,11].

In Central and Eastern Europe, empirical evidence on these relationships remains limited compared with Western Europe, despite substantial sociocultural and educational specificities that influence students’ behavioral repertoires. Recent research from Poland, the Czech Republic, and Slovakia underscores the necessity of regional contextualization in the analysis of physical activity and mental health correlates. Polish findings have demonstrated that students meeting international PA recommendations report higher levels of life satisfaction and overall psychosocial functioning [12,13] regular participation in structured and leisure-time physical activity has been associated with greater subjective vitality and lower perceived stress [14]. Complementary evidence from Slovakia indicates that students who maintained sufficient PA during restrictive pandemic conditions reported more favorable affective balance and general well-being than inactive peers [15].

These results collectively support the hypothesis that the relationship between PA, SB, and psychological well-being is both robust and context-dependent, shaped by cultural, behavioral, and gender-specific moderators.

Despite the growing body of international literature, regionally grounded evidence from Central and Eastern Europe remains scarce, and domain-specific analyses of subjective well-being among university students are largely absent.

Therefore, the present study aimed to examine the associations between physical activity, sedentary behavior, and multidimensional aspects of subjective well-being in a large sample of Slovak university students, with a particular focus on gender-specific and non-linear patterns.

Building on prior theoretical and empirical frameworks, the following hypotheses were formulated:Higher PA levels will be positively associated with subjective well-being across domains of the Bern Questionnaire (BSW/A).Sedentary time will be negatively associated with indicators of well-being, particularly self-esteem and joy of life.The strength and direction of these associations will differ by gender, potentially exhibiting non-linear or threshold effects.

By integrating behavioral epidemiology with cross-cultural mental health research, this study contributes to a more nuanced understanding of the complex interrelations between physical activity, sedentary behavior, and psychological well-being in the Central European academic environment.

## 2. Materials and Methods

### 2.1. Study Design

This is an observational study with a cross-sectional design.

### 2.2. Participants and Recruitment

Data were collected between December 2023 and February 2024 through an online survey distributed via institutional channels (university departments of physical educa-tion, student associations) and social media platforms (Instagram, Facebook). Recruitment followed a convenience sampling approach; thus, the results are representative of this volunteer sample rather than the entire national student population.

A total of 1665 responses were received. Following the IPAQ Research Committee (2005) data-cleaning protocol, 351 incomplete or implausible questionnaires (e.g., total ac-tivity >16 h/day) were excluded. The final analytic sample comprised 1314 participants (69.5% women, 30.5% men; mean age = 20.71 ± 1.41 years) enrolled in 20 public universi-ties across Slovakia. However, because the sample was obtained through convenience sampling via institutional and social media channels and consisted predominantly of women (69.5%), the findings should not be considered nationally representative. The re-sults are generalizable primarily to first-year university students with similar demograph-ic characteristics and voluntary participation patterns.

Prior to the completion of the survey, all participants provided informed consent. Par-ticipation was voluntary and anonymous. The study was approved by the institutional ethics committee and was conducted in accordance with the Declaration of Helsinki. The study protocol was approved by the Human Research Ethics Committee of Pavol Jozef Šafárik University in Košice (approval No. PJSU-05/2023).

### 2.3. Measures

#### 2.3.1. Demographic and Anthropometric Characteristics

Participants reported gender, age, residence, study field, and self-rated health.

Height and weight were self-reported, and Body Mass Index (BMI) was calculated as weight (kg) divided by height squared (m^2^). Self-reported anthropometrics are subject to potential bias (i.e., underestimation of weight and overestimation of height), which was considered in the interpretation of BMI-related findings.

#### 2.3.2. Subjective Well-Being

Subjective well-being was assessed using the Bern Questionnaire on Subjective Well-Being—Adult Form (BSW/A) [16], adapted and validated for the Slovak population by Džuka [17]. The instrument comprises six domains: (A) positive attitude toward life, (B) problems, (C) somatic complaints, (D) self-esteem, (E) depressed mood, and (F) joy of life. Each domain includes four items rated on a 6-point Likert scale (1 = never, 6 = always). Higher scores indicate a greater intensity of the corresponding domain construct. Internal consistency in the present sample was satisfactory (Cronbach’s α = 0.81–0.89 across domains).

#### 2.3.3. Physical Activity

PA was assessed using the International Physical Activity Questionnaire—Short Form (IPAQ-SF), a validated self-report instrument for adults aged 15–69 years. The IPAQ-SF records total weekly minutes of walking, moderate, and vigorous activities performed for at least 10 min at a time. MET-minutes per week were calculated using standard coefficients:Total MET-min/week = (3.3 × walking) + (4.0 × moderate) + (8.0 × vigorous). 

Based on the official IPAQ thresholds, participants were classified into:Low activity (LPA): <600 MET-min/week;Moderate activity (MPA): 600–2999 MET-min/week;High activity (HPA): ≥3000 MET-min/week.

Sedentary behavior was assessed using the IPAQ sitting-time item:
“During the last 7 days, how much time did you usually spend sitting on a typical day?”

Responses were recorded in minutes per day.

Data screening followed the official IPAQ data-cleaning manual [18]. Cases exceeding plausible thresholds (>960 min/week per domain or >16 h/day sitting) were excluded. For missing data, listwise deletion was applied for key variables (PA, well-being), whereas variables with <5% missingness were imputed using series mean substitution to preserve sample size.

#### 2.3.4. Statistical Analysis

Descriptive statistics (mean, SD, median, interquartile range) were used to characterize the sample and distribution of PA and well-being outcomes. Data distribution was evaluated using the Shapiro–Wilk test (α = 0.05), confirming non-normality; therefore, non-parametric tests were used.

Between-gender differences were analyzed using the Mann–Whitney U test, and differences across PA levels were examined using the Kruskal–Wallis H test, with Dunn–Bonferroni post hoc comparisons. Effect sizes were computed as r (rank-biserial correlation) for Mann–Whitney tests, η^2^/ε^2^ and Spearman’s ρ with 95% confidence intervals (CIs).

To address multiple comparisons, the Benjamini–Hochberg False Discovery Rate (FDR) [19] correction was applied; *p*-values reported in tables and text are FDR-adjusted unless specified otherwise. The FDR correction was applied across all comparisons within each well-being domain (six domains × multiple group comparisons), controlling the family-wise error rate at α = 0.05. For all FDR-adjusted analyses, statistical significance was evaluated at q < 0.05. Statistical significance was set at *p* < 0.05 (two-tailed).

All analyses were performed using IBM SPSS Statistics, Version 27.0 (IBM Corp., Armonk, NY, USA) and R, Version 4.2.3 (R Foundation for Statistical Computing, Vienna, Austria), utilizing the packages rstatix, ggplot2, psych, and FSA.

## 3. Results

Gender-stratified analyses using the Mann–Whitney U test revealed significant differences across several BSW/A domains (Table 1). Women scored higher in Problems (B) and Depressed mood (E), whereas men exhibited higher Self-esteem (D) and Positive attitude (A) (all FDR-adjusted *p* < 0.01). Differences in Somatic complaints (C) and Joy of life (F) were small and nonsignificant (*p* > 0.05). Effect sizes ranged from *r* = 0.10 − 0.28, indicating small-to-moderate magnitude differences.

### 3.1. Physical Activity Level and Quality of Life

Kruskal–Wallis tests indicated significant differences across PA categories for Self-esteem, Depressed mood, and Somatic complaints (H(2) = 26.3–32.1, all *p* < 0.001, ε^2^ = 0.06–0.09), whereas no FDR-corrected differences were observed for Problems or Positive attitude (Table 2). Post hoc Dunn–Bonferroni comparisons showed that students in the high PA category reported higher Self-esteem and Joy of life and lower Depressed mood and Somatic complaints than those in the low PA category (adjusted *p* < 0.01). Differences between the moderate and high PA categories were small and mostly nonsignificant, consistent with a pattern in which additional PA does not correspond to further differences once a moderate threshold is reached.

To complement the categorical analyses, gender-stratified correlational analyses were performed using continuous PA values. Among women, continuous PA exhibited a weak inverse association with Self-esteem (r = −0.24, *p* < 0.01). This correlational pattern differs from the categorical comparison, in which moderate-activity women showed slightly higher Self-esteem than low-activity women (Table 2). Differences between these two approaches are methodologically plausible because categorizing continuous PA into broad groups reduces variability and introduces cutpoints that may not align with the underlying distribution. Therefore, the inverse correlation observed among women should be considered with caution, as it may partly reflect methodological differences between categorical and continuous operationalizations of PA.

### 3.2. Correlations Between Physical Activity, Sedentary Behavior, and Well-Being

Gender-stratified correlational analyses are presented in Table 3. Among women, higher total PA showed weak positive associations with Self-esteem and Joy of life and weak negative associations with Depressed mood and Somatic complaints (ρ = 0.13 to −0.19, all FDR-adjusted *p* < 0.05). Sitting time displayed small positive associations with Problems and Somatic complaints, though most associations did not remain significant after FDR adjustment.

Among men, total PA showed weak-to-moderate positive associations with Self-esteem and Joy of life and negative associations with Depressed mood and Somatic complaints (ρ = 0.17 to −0.24, all FDR-adjusted *p* < 0.05). Sitting time showed no significant associations with any well-being domain.

To clarify the pattern noted in Section 3.2, we examined the correlation between continuous moderate-to-vigorous PA and self-esteem among women. The inverse association (ρ = −0.24, FDR-adjusted *p* < 0.01) differs from the categorical comparisons in Table 2, where moderate-activity women reported slightly higher median self-esteem than low-activity women. This discrepancy likely reflects methodological differences: categorizing PA reduces variance and imposes arbitrary cutpoints, which may obscure or distort underlying non-linear associations. The correlational pattern should therefore be interpreted cautiously.

### 3.3. Hypothesis Testing Summary

**Hypothesis** **1.**
*Predicting positive associations between physical activity and well-being across the BSW/A domains, was supported. Higher PA was consistently associated with higher self-esteem and joy of life and with lower depressed mood and somatic complaints.*


**Hypothesis** **2.**
*Predicting negative associations between sedentary time and well-being, was not supported. Correlations between sitting time and well-being were weak and did not remain significant after FDR correction.*


**Hypothesis** **3.**
*Predicting gender-specific and non-linear associations, was partially supported. Gender differences were observed, and an inverse correlation between moderate PA and self-esteem appeared among women. A plateau pattern among men was visible in group comparisons but did not emerge in linear correlations, suggesting a non-linear association rather than a uniform trend.*


## 4. Discussion

This study examined the associations between physical activity (PA), sedentary behavior, and multiple domains of subjective well-being among Slovak university students, with emphasis to gender-specific and non-linear patterns. Higher PA levels were consistently associated with more favorable well-being indicators, particularly higher self-esteem and joy of life and lower depressed mood and somatic complaints. These findings align with observational research reporting similar associations between PA and psychological well-being in university populations [21,22,23]. Sedentary time demonstrated weak and inconsistent associations, a pattern commonly observed when total sitting duration is measured without differentiating activity type.

Gender differences were evident across several well-being domains, with women reporting higher levels of depressed mood, somatic complaints, and problems, and men reporting higher self-esteem and more positive attitudes. These patterns correspond with previous research documenting gender disparities in university mental health [24,25,26]. However, the mechanisms underlying these differences—such as coping strategies, motivational orientations, or body-related factors—were not assessed in this study and cannot be determined. Prior literature suggests that broader contextual and psychosocial factors may shape gendered experiences of PA and well-being [27,28]. underscoring the need for future research using direct measures of motivation, perceived competence, and self-perception.

Two non-linear patterns emerged. Among women, moderate PA showed an inverse association with self-esteem despite group comparisons indicating slightly higher self-esteem in the moderate-activity group relative to the low-activity group. This suggests a complex association that may involve unmeasured contextual factors. Among men, a plateau pattern was observed, with well-being indicators leveling off beyond moderate PA levels. Similar non-linear or inverse patterns have been described in recent research emphasizing the importance of motivational and contextual factors as correlates of PA–well-being associations rather than explanatory mechanisms [29,30,31,32].

Sedentary time was weakly associated with well-being, consistent with evidence indicating that the qualitative context of sedentary behavior may be more relevant than total duration. Cognitively engaging sedentary activities, such as studying or reading, have been linked to more neutral or favorable psychological profiles compared with passive screen-based behaviors [33,34,35]. Overall, these findings reinforce that PA–well-being associations are not uniform but vary across behavioral, psychosocial, and gender-related factors. This highlights the importance of gender-sensitive and contextually tailored approaches in academic settings and the need to interpret findings cautiously given the cross-sectional design.

### 4.1. Limitations

Several limitations should be considered when interpreting the findings of this study.

First, the cross-sectional design does not allow conclusions regarding temporal sequence or directionality, and reverse associations (e.g., individuals with more favorable well-being engaging in higher PA) cannot be excluded. Second, all variables were assessed through self-report instruments, including the IPAQ-SF and BSW/A, which are susceptible to recall bias and potential misestimation of physical activity, sedentary time, and well-being. Although students from all Slovak public universities were invited, convenience sampling via institutional and social media channels and the predominantly female composition of the sample limit representativeness. Baseline levels across several well-being domains were relatively favorable (e.g., median Joy of life 3.70–3.85; median Depressed mood 2.60–3.20 on a 6-point scale), suggesting a psychologically healthy sample. This may have introduced ceiling effects that reduced the ability to detect stronger associations, particularly in high-functioning domains. The study also did not assess potential mechanisms underlying gender-specific and non-linear patterns, such as exercise motivation, perceived competence, or body image concerns, limiting interpretability of findings such as the moderate-PA paradox observed in women. In addition, the IPAQ-SF captures total sitting time only and does not distinguish between cognitively engaging sedentary behaviors (e.g., studying) and passive screen time, which constrains interpretation of the weak sedentary behavior associations and limits the ability to formulate behavior-specific recommendations. The sample consisted exclusively of first-year university students, and it remains unclear whether similar patterns would be observed among students in later academic years, who may differ in stress exposure, routines, and activity patterns. Finally, important potential confounders—such as socioeconomic status, study field, residence, or daily study load—were not controlled for and may have contributed to unexplained variance in the observed associations.

### 4.2. Implications and Future Directions

Despite these limitations, the study offers several important implications.

From a practical perspective, interventions designed to promote student mental health should adopt gender-sensitive approaches that account for differences in motivation, self-perception, and coping styles. University wellness programs should focus on cultivating intrinsic motivation and body-positive exercise environments, which have been linked to stronger and more sustainable psychological benefits. Moreover, strategies to reduce sedentary time should distinguish between active and passive sedentary behaviors, prioritizing the reduction in screen-based leisure while acknowledging that some academic sedentary tasks are cognitively beneficial.

From a theoretical standpoint, the findings reinforce the need for multidimensional and non-linear models in understanding the PA–well-being relationship. Future research should employ longitudinal and mixed-method designs, integrating objective activity measurements (e.g., accelerometers) with qualitative data on motivation and psychosocial context. Examining potential moderators such as social support, perceived competence, and self-concept may help clarify the contextual factors through which physical activity is related to mental health and resilience in university student populations.

## 5. Conclusions

This study found that higher levels of physical activity were consistently associated with more favorable subjective well-being outcomes among Slovak university students, particularly reflected in higher self-esteem, lower depressed mood, and fewer somatic complaints. The observed relationships were non-linear and gender-specific. Men showed a leveling-off of psychological well-being indicators beyond moderate activity levels, whereas women exhibited more complex patterns that may reflect motivational and self-perception factors.

Although these findings align with international evidence linking physical activity with higher mental health, the cross-sectional design limits causal interpretation. Alternative explanations—such as ceiling effects, self-selection, or unmeasured contextual and psychosocial variables—should be considered when interpreting these associations.

Future research should apply longitudinal and mixed-method approaches to clarify causal mechanisms and to incorporate motivational, social, and environmental determinants of activity. Tailored, gender-sensitive approaches that support intrinsic motivation and balanced activity levels are associated with more favorable physical and psychological well-being profiles in university populations.

## Figures and Tables

**Table 1 healthcare-13-03274-t001:** Gender differences in quality-of-life domains (Bern Questionnaire).

Domain	Women Median (Q1–Q3)	Men Median (Q1–Q3)	U	*p*	*r*	Interpretation
A—Positive attitude toward life	3.55 (3.1–3.9)	3.80 (3.4–4.1)	142,335.5	<0.001 ***	0.21	Small–moderate difference
B—Problems	2.82 (2.3–3.3)	2.60 (2.1–3.1)	151,942.0	0.006 **	0.14	Small difference
C—Somatic complaints	2.86 (2.4–3.4)	2.79 (2.3–3.2)	179,518.2	0.091	0.07	Negligible difference
D—Self-esteem	2.64 (2.3–3.0)	2.93 (2.5–3.3)	137,114.6	<0.001 ***	0.28	Moderate difference
E—Depressed mood	3.20 (2.7–3.6)	2.90 (2.5–3.3)	146,812.4	0.004 **	0.16	Small–moderate difference
F—Joy of life	3.70 (3.3–4.0)	3.74 (3.3–4.1)	183,275.3	0.412	0.04	No meaningful difference

Note: Values are median (Q1–Q3). Mann–Whitney U tests; *p*-values FDR-adjusted. r = rank-biserial effect size. Interpretation based on Cohen (1988) [20] guidelines: r < 0.10 negligible, 0.10–0.29 small, 0.30–0.49 moderate. *** *p* < 0.001, ** *p* < 0.01.

**Table 2 healthcare-13-03274-t002:** Differences in subjective well-being across physical activity levels.

Domain	LPA Median (Q1–Q3)	MPA Median (Q1–Q3)	HPA Median (Q1–Q3)	H(2)	*p*	Effect Size (ε^2^)	Interpretation
A—Positive attitude	3.45 (3.0–3.9)	3.60 (3.1–4.0)	3.85 (3.3–4.2)	8.72	0.013	0.02	Small difference
B—Problems	2.75 (2.2–3.2)	2.68 (2.1–3.1)	2.60 (2.0–3.0)	3.11	0.190	0.01	Negligible difference
C—Somatic complaints	2.95 (2.5–3.4)	2.75 (2.3–3.3)	2.55 (2.1–3.0)	21.4	<0.001 ***	0.06	Moderate difference
D—Self-esteem	2.61 (2.3–3.0)	2.69 (2.4–3.1)	2.91 (2.6–3.4)	26.3	<0.001 ***	0.07	Moderate difference
E—Depressed mood	3.12 (2.6–3.6)	2.85 (2.4–3.3)	2.60 (2.3–3.1)	32.1	<0.001 ***	0.09	Moderate difference
F—Joy of life	3.55 (3.1–3.9)	3.70 (3.3–4.0)	3.85 (3.4–4.1)	9.02	0.011	0.03	Small difference

Note: Values are median (Q1–Q3) due to non-normal distributions (Shapiro–Wilk test, *p* < 0.05). Kruskal–Wallis tests with Dunn–Bonferroni post hoc comparisons were applied; *p*-values are FDR-adjusted. ε^2^ = effect size based on Cohen’s guidelines: <0.01 negligible, 0.01–0.05 small, 0.06–0.13 moderate. *** *p* < 0.001.

**Table 3 healthcare-13-03274-t003:** Gender-stratified correlations between physical activity, sedentary behavior, and well-being domains (Spearman ρ).

Variable	A— Positive Attitude	B— Problems	C—Somatic Complaints	D—Self-Esteem	E— Depressed Mood	F—Joy of Life	Interpretation
Women (n = 915)							
Total PA	+0.12 ***	−0.07 *	−0.14 ***	+0.15 ***	−0.19 ***	+0.13 ***	Weak correlations
Sitting time	−0.05	+0.08 *	+0.09 *	−0.06	+0.07	+0.06	Mostly non-significant
Men (n = 399)							
Total PA	+0.16 **	−0.05	−0.11 *	+0.21 ***	−0.24 ***	+0.17 ***	Weak to moderate
Sitting time	−0.04	+0.05	+0.08	−0.04	+0.02	+0.03	Non- significant

Note: Spearman’s ρ (two-tailed). Correlations are interpreted according to Cohen [20]: ρ < 0.10 negligible, 0.10–0.29 weak, 0.30–0.49 moderate. * *p* < 0.05, ** *p* < 0.01, *** *p* < 0.001. All *p*-values were adjusted using the Benjamini–Hochberg FDR procedure. Sitting time measured in minutes per day (min/day). PA = Physical activity; SB = Sedentary behavior.

## Data Availability

The raw data supporting the conclusions of this article will be made available by the authors on request.

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
