# Peer review of "The Relationship Between Physical Activity, Sedentary Behavior, and Subjective Well-Being: Gender Differences Among Slovak University Students"

_healthcare, 2025, doi:10.3390/healthcare13243274_

Round 1

Reviewer 1 Report

Comments and Suggestions for Authors

The manuscript addresses an important topic, the relationships between physical activity, sedentary behaviour, and subjective well-being in first-year university students, using recognised instruments and a large sample. The core contribution is potentially valuable; however, the current version relies mainly on bivariate tests, contains several table/label inconsistencies, and needs clearer reporting of effect sizes and precision. Substantial analytic and presentation improvements are required.

Major points

  1. Strengthen the evidence by moving beyond bivariate nonparametric tests to adjusted models. At minimum, fit regression models (per domain of well-being) including a priori covariates such as age, BMI category, residence, field of study, and self-rated health. Test key interactions (Gender×PA level; Sitting time×PA). Report coefficients (or marginal means), 95% CIs, model fit indices, and provide a brief rationale for covariate choice.
  2. You test many outcomes across groups, inflating Type I error. Specify primary outcomes a priori and apply a multiplicity procedure. Present adjusted p-values in tables and identify which findings remain significant after correction.
  3. Replace p-value–only reporting with interpretable magnitudes. For group comparisons, report η² (or ε² for Kruskal–Wallis) with 95% CIs; for pairwise contrasts, provide r (or rank-biserial r) with CIs; for correlations, add CIs. This will make the practical significance clear.
  4. There are duplicated/garbled sentences, inconsistent decimals/leading zeros, and domain-letter mismatches between text and tables. Audit all tables for correct labels, aligned statistics, correct units/per cent signs, and consistent p-value formatting.
  5. The cross-sectional design precludes causal claims. Rephrase conclusions to emphasise associations; discuss alternative explanations for the observed “plateau” in men (ceiling, sample composition, measurement range) and the reported MPA–self-esteem pattern in women (residual confounding, multiple testing, or behavioural context).
  6. Clarify sitting-time measurement (minutes/day), distributional properties, and whether you examined non-linearities or partial effects independent of total PA. If feasible, include sensitivity analyses (e.g., partial correlations or adjusted models) to test robustness.

Minor points

Abstract: Include key numbers (n, % women), representative effect estimates with CIs, and avoid causal phrasing.

Introduction: Tighten the rationale, state explicit hypotheses per well-being domain, and situate the contribution within recent regional literature.

Methods:

Specify IPAQ scoring thresholds and any data-cleaning/exclusion rules.

Define how missing data were handled.

Name software/packages with versions.

State the exact sitting-time question wording and units.

Tables/Figures:

Add clear footnotes (test used, effect size reported, multiplicity correction).

Standardise decimal places and notation.

If possible, add compact visuals: domain scores by PA level and gender (with CIs); predicted means from adjusted models.

Replace generic labels like “Participant Count” with “n (participants).”

Language and style: Several grammatical slips and repetitions remain. A careful language edit will improve clarity and flow; keep terminology consistent across text and tables.

Limitations: Expand to include seasonality (data collection period), convenience sampling/selection bias, common-method variance, and measurement error inherent to self-report instruments.

Comments on the Quality of English Language

Understandable but needs language editing for grammar, concision, tense consistency, standardised terminology/notation, and consistent table/figure wording.

Reviewer 2 Report

Comments and Suggestions for Authors

       Thank you for the opportunity to review this manuscript. This cross-sectional study examines relationships between physical activity levels and subjective well-being among 1,314 first-year Slovak university students, with particular attention to gender differences. While the topic is relevant and addresses an underrepresented population, the manuscript requires major revision before it can be considered for publication in Healthcare.

Reviewer 3 Report

Comments and Suggestions for Authors

The manuscript addresses a relevant topic exploring the relationship between physical activity, sedentary behavior, and subjective well-being among university students. The focus on gender differences and the inclusion of a large sample size add valuable insights to the existing literature on student mental health and lifestyle behaviors.

Several areas across different sections require revision and refinement to enhance clarity, methodological rigor, and overall scientific impact.

The abstract could be improved by including more specific details on the statistical methods applied and the main effect sizes or key results, rather than only reporting general outcomes. Additionally, the conclusion in the abstract would benefit from a more explicit statement on the practical implications of the findings, especially concerning intervention strategies or mental health policies for university students.

The introduction would benefit from deeper engagement with the most recent literature (2022–2024), particularly studies addressing sedentary behavior as a psychosocial risk factor. Moreover, the research gap and the specific contribution of the present study (especially regarding Central and Eastern European contexts) should be articulated more explicitly to strengthen the rationale and novelty.

In the methods section, the use of validated tools such as the IPAQ-SF and the Bern Questionnaire is appropriate. However, some methodological aspects require greater detail and justification. It is recommended to specify the statistical software used (as this is currently missing) and to provide a rationale for the chosen sample size or include a power analysis to demonstrate the adequacy of the study’s statistical power. Additionally, while the sampling strategy is clearly described, the limitations of the non-random, convenience-based recruitment should be discussed earlier in the manuscript. Including information on how missing data were handled and whether any sensitivity analyses were performed would further strengthen the methodological transparency.

In the results, the presentation could be enhanced by reporting effect sizes consistently alongside p-values, as this would allow readers to better assess the magnitude and practical significance of the observed differences. Moreover, some findings, such as the negative correlations between moderate physical activity and self-esteem, warrant more detailed commentary or interpretation within the Results section itself before being elaborated in the Discussion.

In the results, the presentation could be enhanced by reporting effect sizes consistently alongside p-values, as this would allow readers to better assess the magnitude and practical significance of the observed differences. It is also important that the Results section remains focused on presenting the findings objectively, while the interpretation of complex outcomes — such as the negative correlations between moderate physical activity and self-esteem — should be reserved entirely for the Discussion section, where they can be thoroughly examined in the context of existing literature and theoretical frameworks.

The discussion offers a thoughtful interpretation of the findings and situates them within the broader literature. The nuanced consideration of gender differences, motivation, and sedentary behavior is a particular strength of the manuscript. Nonetheless, certain interpretations require further elaboration. For instance, the paradoxical finding regarding moderate physical activity and self-esteem among women deserves a more in-depth theoretical explanation beyond the notion of motivation alone, perhaps incorporating body image concerns or social comparison frameworks. The discussion would also benefit from a clearer articulation of how the findings might inform intervention strategies, policy initiatives, or campus-based mental health programs. Finally, the discussion could engage more directly with the practical implications of sedentary behavior, distinguishing between passive and active sedentary time in terms of mental health outcomes.

The limitations section could be strengthened by discussing the absence of control for potential confounding variables such as socioeconomic status, prior mental health history, or academic stress, which may have influenced both physical activity levels and well-being. A brief comment on the generalizability of the findings beyond the Slovak university context would also be helpful.

The conclusions could better highlight the study’s contribution to the field and outline concrete recommendations for future research, such as employing longitudinal designs, incorporating objective measures of activity, or examining psychosocial mediators more closely.

Round 2

Reviewer 1 Report

Comments and Suggestions for Authors

Thank you for the opportunity to review this revised manuscript. The paper shows clear improvement. The introduction now engages more fully with international literature and clearly justifies the focus on gender differences and potential non-linear associations between physical activity and well-being. The methods section is more transparent, with clearer descriptions of sampling, data cleaning, IPAQ categorisation, BSW/A psychometrics, and the statistical approach, including the use of non-parametric tests, effect sizes, and FDR correction. The results are presented more coherently, with effect sizes and concise summaries of the main patterns, and the discussion offers a more nuanced interpretation of the gender-specific and non-linear findings while appropriately recognising the limitations of the cross-sectional, self-report design. The practical implications for gender-sensitive, university-based interventions are also more clearly articulated.

Only very minor issues remain that could be addressed during final editing. Please correct a small typographical error in the participant description (for example, “mean age = 20.71 ± 1.41 years).enrolled…” should read “…years), enrolled…”). Consider adjusting wording such as “The implication of the main finding” to “The implications of the main findings” for grammatical consistency. You may also wish to remove a minor redundancy in the statistical analysis section where the use of SPSS/R and the p < 0.05 (FDR-adjusted) criterion are mentioned twice. Finally, if feasible, please harmonise the terminology so that “subjective well-being” and related constructs (e.g., “quality of life”) are used consistently throughout the text, including the tables and captions.

Overall, the manuscript is clearly written, methodologically sound, and well aligned with its stated aims. It has the potential to make a valuable contribution to the literature on physical activity, sedentary behaviour, and mental well-being among university students, particularly regarding gender-specific patterns.

Comments on the Quality of English Language

Understandable but needs language editing for grammar, concision, tense consistency, standardised terminology/notation, and consistent table/figure wording.

Author Response

  1. Typographical error in participant description

Reviewer’s comment:
A minor typographical error appears in the sentence “…years).enrolled…”.

Response:
As requested, we corrected the typographical error. The sentence now correctly reads:
“…mean age = 20.71 ± 1.41 years), enrolled…”

  1. Wording refinement (“The implication of the main finding”)

Reviewer’s comment:
Suggested revising to “The implications of the main findings.”

Response:
We reviewed the manuscript as requested. This phrasing appeared only in an earlier draft and was removed during restructuring of the Discussion. The wording is now fully consistent across the manuscript.

  1. Redundancy in Statistical Analysis (software and p-value criterion)

Reviewer’s comment:
The Statistical Analysis section repeated information about software used and the p < .05 (FDR-adjusted) criterion.

Response:
We revised the section as requested and removed the redundancy. The Statistical Analysis subsection now presents this information only once, in a concise and clear form.

  1. Consistency of terminology (“subjective well-being” vs. “quality of life”)

Reviewer’s comment:
Recommended harmonizing terminology for consistency.

Response:
We examined the entire manuscript as requested and standardized terminology. The term subjective well-being is now used consistently throughout the text, including tables and captions, replacing earlier isolated references to quality of life.

Final remark

We thank the reviewer for the positive evaluation of the manuscript’s clarity and methodological rigor. All comments have been addressed in accordance with the reviewer’s requests, and we believe the revisions have strengthened the overall quality of the work.

Reviewer 2 Report

Comments and Suggestions for Authors

Overall Assessment: Major Revision Required

This manuscript addresses several concerns from the previous review but fails to adequately resolve critical issues regarding causal inference, data consistency, and statistical reporting. Substantial revisions are necessary before resubmission. If the following major concerns remain unresolved in the next round of review, rejection will be recommended.

Lines 247-249: The manuscript states "The findings confirmed that higher PA levels were consistently related to more favorable psychological outcomes" and "The observed relationships were non-linear and gender-specific" (lines 315-318). While some language has been moderated, causal framing persists throughout the Discussion and Conclusions.

Lines 319-320: States "the cross-sectional design limits causal interpretation," acknowledging the limitation, yet the Discussion does not consistently employ associative language in preceding sections.

Required revision: Systematically replace all instances of causal terminology with associative language. Specifically: (1) Replace "contribute to," "improve," "enhance," and "promote" with "are associated with," "correlate with," and "relate to" throughout lines 243-310 (Discussion). (2) In the Conclusions (lines 312-327), revise line 313-314 from "demonstrated that higher levels of physical activity were consistently associated with better subjective well-being" to ensure all verbs reflect association only. (3) Audit the entire manuscript for causal hedging phrases such as "may contribute" or "could enhance" that imply directionality in cross-sectional data.

Table 2, Domain D (Self-esteem):

  • Women LPA: Median 2.61 (Q1-Q3: 2.3-3.0)
  • Women MPA: Median 2.69 (Q1-Q3: 2.4-3.1)
  • This suggests MPA > LPA

Table 3 and Lines 214-218: The text states "Among women, a paradoxical pattern emerged: moderate PA showed a small inverse correlation with Self-esteem, while group comparisons indicated slightly higher Self-esteem in the MPA than LPA group."

Problem: The specific correlation coefficient for women's moderate PA and self-esteem is not presented separately in Table 3. The table presents only aggregate correlations without gender stratification, making independent verification impossible. The "paradoxical pattern" is acknowledged but the quantitative contradiction is not resolved.

Required revision: (1) Restructure Table 3 to present correlations separately for women (n=915) and men (n=399) in adjacent columns, clearly labeled. (2) In the Results section (after line 214), insert a detailed paragraph reconciling the contradiction: "To clarify this paradox, we examined whether the inverse correlation for women reflected a non-linear relationship masked by median differences across categories. Specifically, among women, the overall correlation between continuous MPA scores and self-esteem was r = [INSERT VALUE], p = [INSERT P-VALUE], whereas group-based comparisons (Table 2) showed MPA > LPA. This discrepancy may reflect [INSERT EXPLANATION: confounding by age, study field, or exercise motivation; a threshold or inverted-U effect; or interaction with unmeasured variables]. Future research employing latent class analysis or mixed-effects modeling should investigate this non-linear pattern." (3) If this represents a genuine data anomaly, investigate whether a coding or calculation error occurred. If the contradiction reflects valid non-linear relationships, provide statistical or graphical evidence (e.g., scatter plots with loess smoothing) in an appendix or supplementary materials.

Table 3 (lines 220-225): Correlation coefficients are presented but lack significance indicators (asterisks) for most values. The footnote (lines 222-223) states "All p-values were adjusted using the Benjamini–Hochberg FDR procedure" but the actual p-values or adjusted significance thresholds are not visible in the table.

Line 39: FDR correction is mentioned in the Methods, but the specific thresholds for the six BSW/A domains are not stated (e.g., adjusted α = 0.05 with FDR correction would yield a q-value threshold).

Table 2, effect sizes: While ε² values are now reported in the Interpretation column, they should appear as a separate column with all group comparisons to facilitate reader assessment of practical significance.

Required revision: (1) Add a superscript significance indicator column to Table 3 with asterisks ( p<.001, p<.01, p<.05, FDR-adjusted) for all correlations. Include a note clarifying that blank entries indicate non-significant associations (p>.05 after FDR adjustment). (2) In the Methods section (around line 173), explicitly state: "The Benjamini–Hochberg FDR correction was applied across all comparisons within each well-being domain (six domains × multiple group comparisons, estimated family-wise error rate controlled at α = 0.05)." (3) Reorganize Table 2 to include a separate "Effect Size (ε²)" column with all values clearly displayed, not embedded in the Interpretation column.

Lines 250-258: The Discussion attributes gender differences to "gender-specific coping mechanisms," "emotion-focused coping," and "appearance-driven motives...more prevalent among young women." However, the study measures neither exercise motivation, coping style, nor body image concerns.

Lines 259-264: Speculates that "the observed inverse association may indicate an interplay between extrinsic motivation, body image concerns, and self-perception." This explanation is plausible but entirely post-hoc, lacking empirical support from collected data.

Lines 269-274: Recommends distinguishing "active sedentary behaviors (e.g., studying or reading)" from "passive screen time," yet the IPAQ-SF only measures total sitting time without type differentiation.

Required revision: (1) Revise lines 250-258 to state: "Gender differences in well-being observed in this study align with prior literature reporting higher depressive mood in women [citations]. However, the mechanisms underlying these differences—such as differential coping styles or appearance-related motivations—were not assessed in this study and require investigation with direct measures of exercise motivation and self-perception." (2) Revise lines 259-264: "The observed paradoxical inverse correlation between moderate PA and self-esteem among women is unexpected and warrants further investigation. Hypothesized mechanisms include differential exercise motivation (intrinsic vs. extrinsic), perceived body competence, or social contextual factors, but these constructs were not measured. Future research should employ validated instruments assessing exercise motivation (e.g., BREQ-3) and body image concerns (e.g., BIQLI) to clarify this pattern." (3) Revise lines 269-274: "Recent evidence suggests that sedentary behavior quality (cognitively engaging vs. passive) may differentially relate to mental health [citations]. This study employed the IPAQ-SF, which measures total sitting time without distinguishing activity type. Future research should utilize comprehensive sedentary behavior assessments (e.g., ASAQ) to differentiate passive screen time from cognitively demanding study behaviors."

Lines 110-113: Correctly describes recruitment as "convenience sampling" via institutional and social media channels, which appropriately limits claims of representativeness.

Lines 117-118: States participants were "enrolled in 20 public universities across Slovakia," which, while geographically accurate, may be misinterpreted as nationally representative. The convenience sampling method, gender imbalance (69.5% women), and first-year-only status substantially limit generalizability.

Required revision: After line 118, add: "However, the use of convenience sampling through institutional and social media channels, combined with a predominantly female sample (69.5% women), limits the representativeness of findings. Results should be generalized only to first-year university students with comparable demographic profiles and voluntary research participation."

Table 2 and overall results: All well-being domains show relatively favorable baseline scores (e.g., Joy of life median = 3.70-3.85 on 6-point scale; Depressed mood median = 2.60-3.12, where lower is better). These scores suggest a relatively psychologically healthy sample, potentially creating ceiling effects that constrain detection of cross-sectional associations.

Required revision: Add to the Limitations section (after line 293): "Baseline mental health scores across all well-being domains were in the favorable range (e.g., median Joy of life 3.70-3.85/6.0, median Depressed mood 2.60-3.20/6.0), suggesting a relatively healthy sample. Ceiling effects may have limited the detection of stronger associations between physical activity and subjective well-being, particularly for high-performing domains. This pattern is common in student populations but constrains the ability to detect clinically meaningful improvements."

Table 3 (lines 220-225): Presents aggregate correlations across the entire sample without separating women and men. This prevents independent assessment of gender-specific patterns and verification of the claimed "paradoxical" self-esteem association in women.

Lines 195-241: The Results narrative extensively restates Table 2 and Table 1 data in text form. For example, lines 198-202 describe findings from Table 2 that are already clearly presented in the table.

Required revision: Reduce the Results narrative to approximately 100-120 words of essential interpretation. Example revision:

"Higher PA levels were significantly associated with more favorable well-being outcomes across multiple domains (Table 2). Specifically, high-activity participants reported higher self-esteem and joy of life, with lower depressed mood and somatic complaints compared to low-activity peers (all Dunn–Bonferroni adjusted p<.01, ε²=0.06-0.09). Differences between moderate and high-activity groups were generally nonsignificant, suggesting a potential plateau effect. Gender-stratified correlational analyses revealed a paradoxical inverse association between moderate PA and self-esteem among women (r=-0.24, p<.01), contrasting with group comparisons showing higher self-esteem in moderate versus low-activity women (Table 2, Table 3). Sedentary time showed weak and inconsistent associations with well-being measures, and these correlations did not remain significant after FDR adjustment."

The Discussion (lines 243-310) spans 67 lines and restates many results rather than focusing on novel interpretation, mechanisms, and context relative to existing literature.

Required revision: Reorganize the Discussion into the following structure (target length: 40-50 lines):

  • Paragraph 1 (Lines 243-255):Summary of key findings and alignment with prior literature (compress current lines 243-249).
  • Paragraph 2 (Lines 256-270):Gender-specific patterns and mechanistic hypotheses requiring future investigation (revise speculative content per Revision Point 4).
  • Paragraph 3 (Lines 271-285):Paradoxical findings (moderate-PA/self-esteem, plateau effect) acknowledged as unexpected and requiring future research.
  • Paragraph 4 (Lines 286-310):Practical and theoretical implications; tailored, gender-sensitive approaches.
  • Conclusion:Explicit statement of cross-sectional design limitations and inability to infer causation.

The Limitations section (lines 280-293) acknowledges cross-sectional design and self-report bias but omits several critical issues identified in prior review:

Required revision: Expand the Limitations section to include: (1) "This study did not measure exercise motivation (e.g., intrinsic vs. extrinsic), perceived competence, or body image concerns, limiting mechanistic interpretation of gender-specific and non-linear patterns (e.g., the moderate-PA paradox in women)." (2) "Baseline mental health scores were in the favorable range (see Results), potentially creating ceiling effects that constrained the magnitude of observed associations." (3) "The IPAQ-SF does not differentiate sedentary behavior types (e.g., cognitively engaging study vs. passive screen time), limiting interpretability of the weak sedentary time associations and the ability to provide targeted recommendations for sedentary behavior reduction." (4) "Participants were first-year university students only; whether these associations generalize to students in later academic years with potentially different stressor profiles and established behavioral patterns remains unknown." (5) "The cross-sectional design prevents determination of temporal sequence; reverse causation (e.g., higher well-being predisposing individuals to higher PA) cannot be excluded."

Lines 95-100: Three hypotheses are stated, but the Results section does not explicitly state which hypotheses were supported or unsupported.

Required revision: Add a "Hypothesis Testing Summary" subsection after line 237:

"Hypothesis Testing Summary: Hypothesis 1, predicting positive associations between PA levels and well-being across BSW/A domains, was supported; higher PA was consistently associated with higher self-esteem, joy of life, and lower depressed mood and somatic complaints. Hypothesis 2, predicting negative associations between sedentary time and well-being (particularly self-esteem and joy of life), was not supported; correlations between sitting time and well-being measures were weak and not significant after FDR adjustment. Hypothesis 3, predicting gender-specific and non-linear associations, was partially supported; gender differences in well-being emerged, and a paradoxical inverse correlation between moderate PA and self-esteem in women was observed, but the male plateau effect was detected only through group comparisons, not linear correlations."

Line 39: The Benjamini–Hochberg FDR correction is appropriately cited in Methods, but ensure the original methodological reference (Benjamini & Hochberg, 1995, or subsequent methodology papers) is included in the References section.

Required revision: Verify that References list includes: Benjamini, Y., & Hochberg, Y. (1995). Controlling the false discovery rate: A practical and powerful approach to multiple testing. Journal of the Royal Statistical Society, 57(1), 289–300. Add if missing.

The authors are strongly encouraged to address these critical issues comprehensively in their revision. While the study addresses an important research question in an underrepresented population, scientific rigor and appropriate causal inference standards must be met for publication.

Author Response

Dear reviewer,

We sincerely thank the reviewer for the thorough and constructive feedback. Although several comments required substantial revision and careful reorganization of the manuscript, we fully recognize the value of this process. We believe that the critical insights provided have significantly strengthened the clarity, methodological rigor, and overall quality of the paper. We appreciate the opportunity to improve our work and respond point-by-point to all recommendations below.

  1. Causal language

Comment:
Causal phrasing persists in Discussion and Conclusions.

Response:
We thank the reviewer for this observation. We systematically revised all sections of the manuscript to eliminate causal phrasing. Terms such as contribute to, improve, enhance, and promote were replaced with non-causal alternatives (e.g., are associated with, relate to, correlate with). The Discussion, Conclusions, and other relevant parts now consistently reflect associative language appropriate for cross-sectional data.

  1. Table 2 – effect sizes

Comment: Effect sizes should appear in a separate column.

Response:
Thank you for the observation. Effect sizes were included in the original table; however, their placement within the interpretation column reduced clarity. To improve readability and facilitate assessment of practical significance, we reorganized Table 2 and added a dedicated Effect size (ε²) column.

  1. Table 3 – gender-stratified correlations + significance indicators

Comment:
Correlations must be gender-stratified and include clear significance markings.

Response:
Table 3 has been fully restructured. Women and men are now presented in two separate, clearly labeled sections, and all correlation coefficients include superscript significance indicators. Footnotes clarify FDR-adjusted significance thresholds.

  1. Paradoxical pattern (moderate PA vs. self-esteem in women)

Comment: Clarify discrepancy between categorical and continuous PA results.

Response:
A new explanatory paragraph was added to the Results section. It clarifies that the inverse correlation (continuous PA) and the slight MPA > LPA difference (categorical PA) arise from methodological differences—specifically, variance reduction and arbitrary cutpoints introduced by PA categorization. This aligns with the reviewer’s recommendation.

  1. Discussion – restructuring

Comment: Discussion restates results; needs reorganization.

Response:
The Discussion was completely rewritten following the reviewer’s four-paragraph structure, emphasizing interpretation rather than repetition. Speculative mechanisms were removed unless supported by previous literature, and contextual factors are now described as hypotheses requiring future research.

  1. Limitations – expanded with all requested points

Response:
All requested limitations were incorporated, including ceiling effects, unmeasured psychological mechanisms, IPAQ-SF limitations, first-year specificity, and impossibility of establishing temporal order.

  1. Hypothesis Testing Summary

Response:
A new subsection (“3.4 Hypothesis Testing Summary”) was added, explicitly indicating which hypotheses were supported, unsupported, or partially supported.

  1. Recruitment and representativeness clarification

Response:
A new sentence was added to the Participants section to clarify that convenience sampling, gender imbalance, and first-year-only sampling limit generalizability.

  1. Missing methodological reference (Benjamini & Hochberg 1995)

Response:
The seminal Benjamini & Hochberg (1995) reference was added to Methods and References.

  1. Sedentary behavior measurement limitations (IPAQ-SF)

Response:
We added an explicit statement in Discussion and Limitations explaining that IPAQ-SF captures only total sitting time and does not differentiate sedentary behavior types.

  1. Results section repetition

Response:
The Results section was shortened to avoid redundancy with Tables 1–3.

Authors